# Self-regulation of socioemotional behavior in twin adolescents: Structural validation of a multidimensional inventory

Lea Pulkkinen[1], Asko Tolvanen[1], Stephanie Zellers[2], Jaakko Kaprio[2], Richard J. Rose[3], Alyce M. Whipp[2]*

**1** Department of Psychology, University of Jyväskylä, Jyväskylä, Finland, **2** Institute for Molecular Medicine Finland (FIMM), HiLIFE, University of Helsinki, Helsinki, Finland, **3** Department of Psychological and Brain Sciences, Indiana University, Bloomington, Indiana, United States of America

\* alyce.whipp@helsinki.fi

## Abstract

Instruments for rating socioemotional behavior with a strong theoretical basis, broad coverage of behaviors, and adequate validation are rare. Here, the Multidimensional Peer Nomination Inventory (MPNI) Form SERI (Socio(E)motional Regulation Inventory) was employed in a longitudinal population-based sample of Finnish twins at age 17 to study: (1) the structure of self-ratings on self-regulation of socioemotional behavior, (2) construct, criterion, concurrent, and predictive validity of the scales, as well as invariance analysis, and (3) genetic and environmental factors contributing to individual differences in self-regulation. A bipolar factor for low versus high self-regulation was interpreted as representing vulnerability to a p-factor (general psychopathology) versus src-factor (self-regulation capacity), and respective scales were formed for both. Behavioral regulation in each was further categorized into a Problem behavior scale (comprising subscales for Hyperactive/Inattentive, Aggressive, and Anxious behavior) and a Prosocial behavior scale. Self-ratings on these scales had high correlations with co-twin ratings at age 17 and similarly formed scales for self- and co-twin ratings at age 14. Twin modeling indicated that the p-factor and src-factor are moderately heritable and attributable to both genetic and unique environmental influences. The inventory can be used for self- and sibling ratings in population studies.

## Introduction

Careful assessment of socioemotional behavior in children and adolescents is important in research and clinical work but finding measures for this purpose with adequate coverage of content, solid theoretical basis, and established predictive validity is challenging. One such measure is an inventory (Multidimensional Peer Nomination Inventory, MPNI) used in the longitudinal study of health and social behavior in

**Data availability statement:** The FinnTwin12 data is not publicly available due to the restrictions of informed consent. However, the FinnTwin12 data is available through the Institute for Molecular Medicine Finland (FIMM) Data Access Committee (DAC) ([fimm-dac@helsinki.fi]()) for authorized researchers who have IRB/ethics approval and an institutionally approved study plan. To ensure the protection of privacy and compliance with national data protection legislation, a data use/transfer agreement is needed, the content and specific clauses of which will depend on the nature of the requested data.

**Funding:** Data collection was supported by the National Institute on Alcohol Abuse and Alcoholism (AA-09203, AA-00145) and the Academy of Finland, Finnish Centre of Excellence Programme no. 40166, and the Academy of Finland (grants 100499, 205585, and 264146 to JK). The preparation of the present paper was supported through grants from the NIH (Rutgers grant # R01AA015416 to Jessica Salvatore) and the Academy of Finland Centre of Excellence in Complex Disease Genetics (grant 336823 to JK). This project has also received funding from the European Union's Horizon 2020 research and innovation programme under grant agreement No 874724 (Equal-Life). Equal-Life is part of the European Human Exposome Network. No funding source played any part in the design, analysis, interpretation, or writing of this manuscript.

**Competing interests:** The authors have declared that no competing interests exist.

Finnish twins with multiple informants from age 11/12–17 [1,2]. This paper describes the theoretical basis and structure of the MPNI used with self-ratings at age 17, and genetic and environmental influences affecting assessed behavior.

Research on socioemotional behavior asking how an individual behaves in a particular situation toward others or oneself tends to be deficit-oriented, as identified in the systematic review of 17 measures by Tsang et al. [3]. An exception was the Strengths and Difficulties Questionnaire (SDQ) developed by Goodman [4]. The MPNI, not included in the review by Tsang et al. [3], also covers both strengths and difficulties. The roots of the MPNI extend to the 1960s. It is based on a two-dimensional impulse control model that Pulkkinen, then [5], had devised for the analysis of children's differences in socioemotional behavior. In this model, the horizontal axis depicted the amount of cognitive control over emotional behavior, and the vertical axis the expression or inhibition of behavior (overt activity versus passivity). According to the model, low cognitive control of behavior may appear as uncontrolled expression of impulses, such as aggression (often called externalizing problems), and as uncontrolled inhibition of impulses such as anxiety (often called internalizing problems). Likewise, high cognitive control of behavior may appear in the controlled expression of impulses, such as constructive problem solving, or in the controlled inhibition of impulses, such as compliance.

Compared to other models, Pulkkinen's theoretical reasoning contains assumptions typical of the dual process model, according to which, behavior is jointly produced by two independent systems: the impulsive system and control system [6]. An interest in individual differences also connects the impulse control model to the trait models of impulse control, particularly to the temperament model of Rothbart and Bates [7]. To harmonize her theoretical construct with work of other researchers, Pulkkinen relabeled the concept of cognitive control of behavior first as self-control and later as self-regulation, covering the components for emotion regulation and behavior regulation (see [8]). The terms self-control and self-regulation have been used interchangeably in research literature. Inzlicht et al. [9] suggest, however, that these concepts refer to distinct processes; both concepts "resemble but are not isomorphic with cognitive control" (p. 321). Self-regulation involves steering one's behavior toward a desired end state, and it subsumes regulation not only of behavior, but also of thoughts and emotions. Self-control refers to means of resolving conflicts between competing goals. Cognitive control, in turn, refers to executive functions, such as inhibition, attentional shifting, and working memory [10,11].

The two-dimensional impulse control model, formed by self-regulation and social activity, was validated in the Jyväskylä Longitudinal Study of Personality and Social Development (JYLS) that began with study of Finnish children's (N = 369, 52% males, b. 1959) aggressive and non-aggressive behaviors at age 8 in 1968, since continued through ages 14, 20, 27, 36, and 42 until age 50 [5,8]. Socioemotional behavior was assessed at age 8 by peer nominations and teacher ratings with a 38-item inventory. The JYLS inventory was later adapted for a population-based cohort study of the behavioral development and health of Finnish twins (FinnTwin12). This modified 37-item inventory was applied to peer nomination when twins were 11/12 years old,

and it was called a Multidimensional Peer Nomination Inventory (MPNI). Its Teacher, Parent, Self, and Co-twin Rating forms were also used at age 11/12 and/or 14 [1]. For data collection at age 17, some items of the MPNI were modified for self-rating due to developmental differences and other considerations. The age 17 MPNI was validated in the present study.

Our focus was on the self-regulation dimension of the two-dimensional impulse control model, in other words, on the self-regulation of social and emotional (shortly: socioemotional) behavior that has been found central to developmental psychopathology [11]. The comorbidity of externalizing and internalizing problems has been demonstrated in children's high distress [12], recurrent pain [13], and in highly aggressive behavior [14] that is negatively associated with prosocial behavior [15]. Furthermore, low self-regulation evident in both externalizing and internalizing problems predicts various aspects of social functioning problems in adulthood including long-term unemployment, heavier drinking, and antisocial behavior, as shown by a longitudinal study from childhood to middle age [8]. In contrast, high self-regulation predicted positive social functioning such as higher success at school, occupational attainment and income, refraining from substance use, and non-offending.

The first aim was to analyze the structure of the MPNI self-ratings at age 17 with the expectation that a bipolar factor explains the variance of self-regulation for both problem behaviors and prosocial behaviors (Fig 1). Both low and high self-regulation were expected to be displayed by different behaviors: by problem behaviors such as hyperactivity, aggression and anxiety for low self-regulation, and by prosocial behaviors such as constructiveness and compliance for high self-regulation. Low self-regulation might indicate vulnerability to a general psychopathology (p) factor that has been introduced to denote a common underlying factor that contributes to the development of various forms of mental disorders [16,17], even in early childhood [18]. High self-regulation was, correspondingly, expected to indicate a general self-regulatory capacity (src) that contributes to successful development [8].

The second aim was to investigate different types of validity of self-rating scales at age 17: (i) construct validity regarding low and high self-regulation on the scale level, (ii) criterion validity with indicators of individuals' social positions among their peers as criteria, (iii) concurrent validity with co-twin ratings as criteria, and (iv) predictive validity of corresponding ratings by different informants across development (age groups 11/12 and 14).

The third aim was to study correlations of the scales in monozygotic (MZ) and dizygotic (DZ) twin pairs to estimate genetic and environmental factors affecting self-ratings. Genetic and environmental factors were expected to contribute

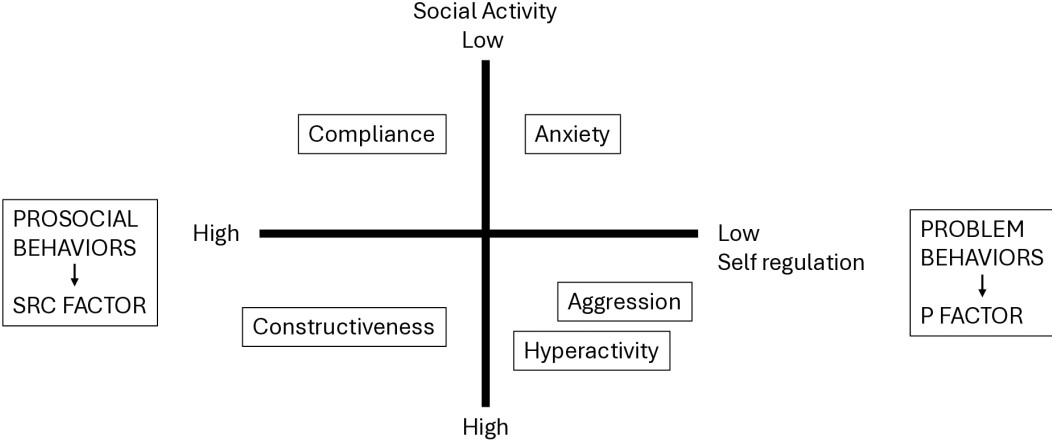

**Fig 1. The expected structure of the MPNI self-ratings at age 17 within the framework of an impulse control model (see [8], p. 8).**

to both low and high self-regulation. At age 11/12, both genetic and environmental effects were found to be significant for different types of socioemotional behavior [19].

## Materials & methods

### Participants

The study employed the FinnTwin12 cohort, a longitudinal population-based sample of Finnish twins born 1983–1987 whose parents were identified in the Finnish Central Population Registry and consented to participate when the twins were 11–12 years of age [2,20]. Data collection began with a family questionnaire (87% participation rate, N = 5600 twins). These families/twins (called, the epidemiologic group) had further questionnaire data collected at age 12 (range: 11.4-12.4; 4920 valid responses), age 14 (range: 13.9-14.9; 4523 valid responses), and age 17 (range: 17.2-19.5; 4041 valid responses) [21]. Additionally, at age 14 a subset of twin families (N = 1035) was asked to participate in a more intensive study. Most of this intensive group was selected at random, but the sample was enriched with twins at elevated familial risk for alcoholism (assessed from a parent questionnaire at the age 12 data collection). Data collected among this intensive group included MPNI self- and co-twin ratings at age 14.

Classification of same-sex twin pairs as MZ or DZ was based on baseline questionnaire responses, separately completed by the twins and their parents, inquiring about the twins' similarity of appearance and early development and how often the twin's appearance led others to confuse their identity. Validation of using DNA was performed in a sample of 295 same-sex twin pairs; zygosity was confirmed among 97% of the pairs (see [21,22]). The present study consisted of 1383 MZ, 1321 same-sex DZ, and 1297 opposite-sex DZ twins.

Ethical approval for all data collection waves was obtained from the ethical committee of the Helsinki and Uusimaa University Hospital District and Indiana University's Institutional Review Board. All data collection and sampling were performed in compliance with the ethical guidelines. In the 1990s, data was collected from teachers with the approval of school authorities. Parents provided consent for twins at ages 11/12 and 14, while twins themselves provided written consent at age 17.

### Measures

The MPNI was originally developed for peer nomination and subsequently adapted for parent, teacher, self, and co-twin ratings. In the epidemiological sample, the ratings were collected at age 11/12 from parents and teachers, at age 14 from teachers, and at age 17 from participants themselves and their co-twins. In the intensively studied sub-sample, ratings were collected from participant twins and their co-twins at age 14. Of note, in the Finnish school system, both teachers and school environments change from age 12–14 for all students when they move from lower to upper elementary school.

The MPNI forms used at ages 11/12 and 14 comprised 37 items that were divided into several subscales obtained from age 12 MPNI factor analyses [1]. The age 17 MPNI consisted of corresponding items (as shown in S1 Text) with a difference that items for depression were not represented because the General Behavior Inventory (GBI) was also included in the age 17 data collection wave. Items represented the scales obtained at age 12 as follows: hyperactivity-impulsivity, aggression, inattention, social anxiety, constructiveness, and compliance.

Two items for emotion regulation, assessing effortful control [23], were added to the age 17 MPNI as the reference variables for the expected self-regulation factor: "I'm reliable and stable. I keep my composure in all situations" and "My moods change often, and I lose my temper easily". Two items were also available as reference variables for a social activity factor: "I'm very energetic, always on the go and often have contact with other people" and "I am quiet, withdrawn, and often alone".

For social position, three items were available: "I'm popular among other youths", "I am often teased", and "I am a good leader", and one item for assessing resilience "It takes me an unusually long time to get over unpleasant events" (Reverse coded). The response scale for each item ranged from 0 (does not apply) to 3 (applies very well).

## Analysis

### Factor analysis

The structure of the age 17 MPNI Self-rating Form was tentatively studied with a series of factor analyses and orthogonal and oblimin rotations to allow correlated factors, both collapsed across sex and stratified by sex. First, all 37 items of the MPNI were included in the factor analysis to test whether the common variance of the variables was explained by low and high self-regulation and social activity versus passivity. Two factors (rotated orthogonally) explained 79% of the common variance (45% F I and 34% F II), other factors were more specific. In sex-separated factor analyses, the common variance explained by the first two factors was 78% for males and females. The first bipolar factor depicted *low* versus *high self-regulation* and the second bipolar factor depicted *social activity* versus *passivity* (S1 Table). These factors were very similar for females and males, although there were some higher negative loadings toward socially passive behavior in females compared to males.

The items for low and high self-regulation were separately factor analyzed for obtaining possible behavioral components for them. For obtaining possible *behavioral components* for low self-regulation and high self-regulation behavioral items that loaded positively and negatively on the first factor were separately factor analyzed. Three factors from the 19 items for low behavioral self-regulation were extracted and they were interpreted as *Hyperactive*/*Inattentive behavior*, *Aggressive behavior*, and *Anxious behavior.* For high self-regulation (8 items), only one significant factor was extracted. It was interpreted as *Prosocial behavior*. Negative loadings were also expected in the items for giving up and avoidance behavior to depict high self-regulation in a less active way (compliance, see Fig 1), but they did not load onto the factor for high self-regulation in the same way as at age 11/12. Therefore, they were placed in the category "Others" (S1 Table). Also, items depicting sociometric status (popularity, leadership, and victimization) were placed in the category "Others".

After omitting the six items in the category of "Others", the MPNI Form SERI (Socio[E]emotional Regulation Inventory) was formed from the remaining 31 items that were more specifically targeted to the study of self-regulation (see S1 Table for items). Two models were tested for SERI. First, a two exploratory factors model was estimated in which items were specified as categorical, and then a four exploratory factors model was estimated. The estimator used was the WLSMV (weighted least squares mean and variance adjusted) in the Mplus statistical program [24]. Model fits were evaluated using the chi-square test, RMSEA (root mean square error of approximation), CFI (comparative fit index), TLI (Tucker – Lewis Index) and SRMR (standardized root mean square residual). For a good-fitting model, chi-square test values are non-significant, CFI and TLI are near 0.95, and RMSEA and SRMR are below 0.06 and 0.08, respectively [25]. Because twins are more similar than two randomly selected individuals and had been sampled as twin pairs, the standard errors were corrected using the COMPLEX option in Mplus.

### Rating scales

Rating scales were created from self-ratings for each factor obtained by using the mean scores of items identified, allowing 2 missing items for scales with 8 or more items, and only 1 missing item for scales with 4–7 items. Corresponding scales were created from co-twin ratings. Cronbach's alphas for the scales were calculated, separately by sex, to assess the reliability of the scales. Factor analyses and alpha analyses were performed in Stata, version 16 (StataCorp. 2019. College Station, TX, USA).

Corresponding scales were also created from parent ratings at age 12, teacher ratings at ages 12 and 14, and self- and co-twin ratings at age 14. The scales for different informants and ages were formed from the MPNI items corresponding to the scales formed at age 17 self-ratings (see S1 Text for the items in each corresponding scale at age 12 or 14).

Measurement invariance between girls and boys was tested separately for each informant. Then, measurement invariance across informants was tested separately for girls and boys using confirmatory factor analysis, with only those items that theoretically belonged to a factor taken. In all measurement invariance analyses, items were specified as categorical in the model, and the estimator used was the WLSMV in the Mplus statistical program [24].

### Correlations

The validity of self-rating scales was assessed by calculating correlations between scales and with certain criteria. Correlations were calculated using the mean scores of items for each scale. We calculated Spearman correlations due to the items in the scales being ordinal. All correlations were performed in Stata, version 16. To assess construct validity of the self-rating scales at age 17, the intercorrelations of the scales were calculated. To assess their criterion validity, correlations of self-ratings were calculated with indicators of individuals' social positions among their peers at the same age. To assess concurrent validity, correlations between self-ratings and co-twin ratings at age 17 were calculated, and to assess predictive validity correlations of the self-ratings at age 17 were calculated with ratings by different informants across development (age groups 12 and 14). To ensure that we measured the same phenomenon across development, we analyzed measurement invariance at different ages and different informants. We further analyzed measurement invariance between girls and boys.

### Twin analyses

All twin analyses were conducted in R using the package OpenMx [26,27]. First, we calculated within-twin-pair correlations for the scales stratified by sex and zygosity. We estimated within-twin-pair correlations for both self-ratings and co-twin ratings at ages 14 and 17. We also estimated within-twin-pair correlations using a random sub-sample of the age 14 enriched data to identify any possible bias in the twin correlations due to the sampling strategy (S2 Table).

We then reproduced the factor model as a twin model to evaluate the genetic and environmental variance underlying the scales formed and their covariates, as well as their genetic and environmental covariance (Figs 2 and S1). Twin models leverage the amount of genetic variation shared by MZ and DZ twins; MZ twins share 100% of their genetic variation, whereas DZ twins share only 50%, the same as any other pair of full siblings. MZ and DZ twin pairs both share many environmental and other factors (e.g., birth cohort, age, prenatal conditions, socioeconomic status, neighborhood and rearing environment) but also have unique experiences across development. These known similarities within twin pairs allow for the decomposition of trait variance into various sources: additive genetic effects (A), shared environmental influences (C), and unique environmental influences (E). Shared environmental influences are those factors that make twins in a pair more alike, whereas unique environmental influences are those factors that make twins in a pair more dissimilar.

Within the context of the twin factor model, we estimated A, C, and E variance components for the model generated by the factor analysis, as well as 2 covariate items for emotion regulation at age 17. At age 14, items for oppositional behavior and agreeableness approximated low and high emotion regulation, respectively, because the mood lability/stability items were not included in the age 12/14 version of the MPNI. Additionally, we estimated unique A, C, and E variance components for each scale for these factors to understand the variation in scales not explained by the factor. Lastly, we estimated A, C, and E components contributing to the covariation between the factors and covariates. These covariances were transformed for interpretability into phenotypic, genetic, and unique environmental correlations (i.e., rP, rG and rE) between the factors and covariates.

## Results

### Self-regulation and activity as the dimensions of self-rated socioemotional behavior

In total, 4105 twin individuals with self-ratings for the age 17 MPNI Form SERI were included in the factor analysis. The two-factor structure of self-regulation and activity was estimated with the two exploratory factors model using a subset of 31 items, titled the MPNI Form SERI. The first bipolar factor (Table 1) was loaded by the reference variables for emotional self-regulation, positively by labile mood (#21) and negatively by stable mood (#29), indicating effortful control in self-regulation [7]. Positive loadings were accordingly found in items for hyperactive/inattentive, aggressive, and anxious behavior. Likewise, negative loadings were found in items for constructive and compliant behavior. The first bipolar factor

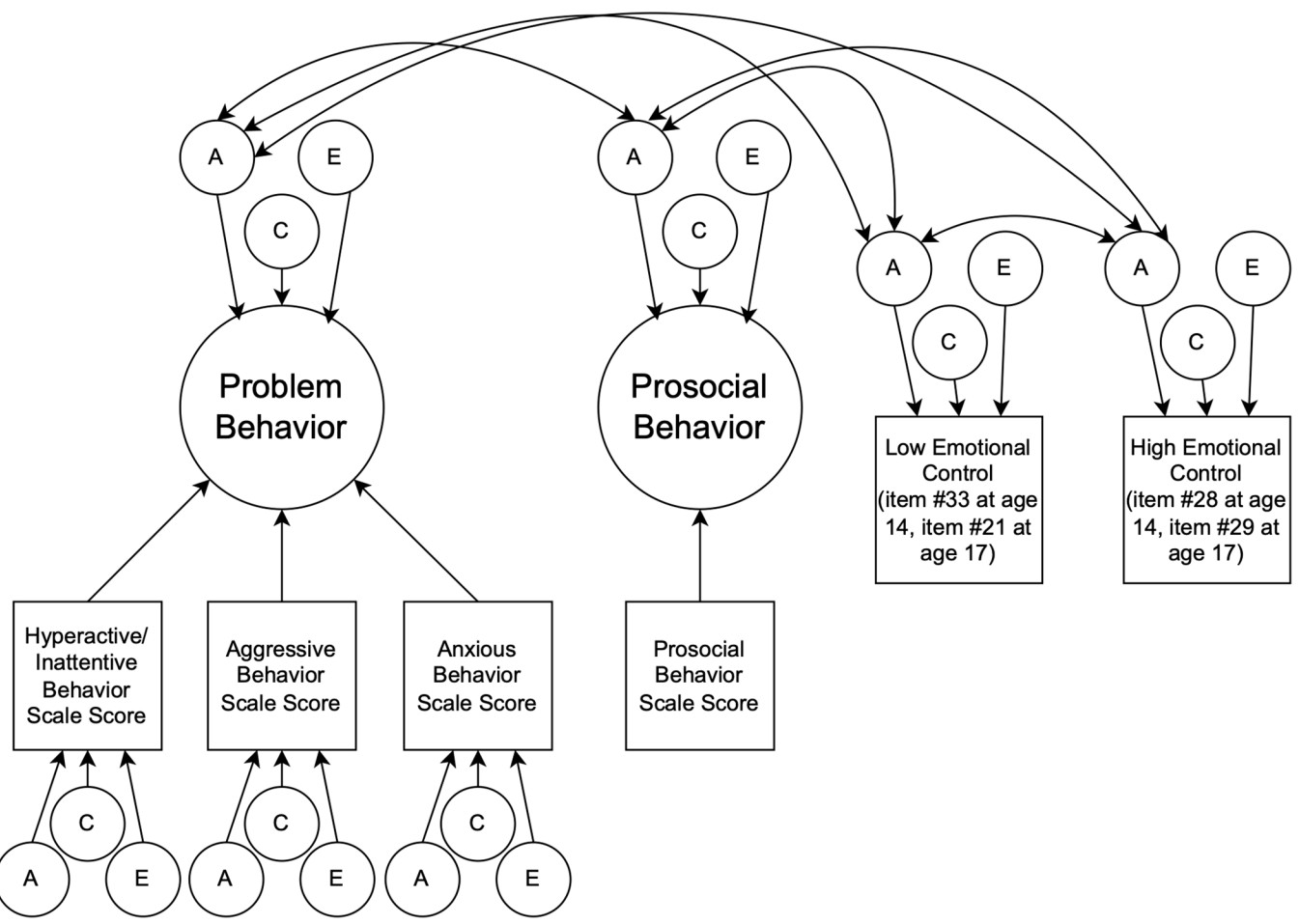

**Fig 2. Path diagram depicting factor model parameterized as a twin model.** Note that not all correlation arrows are depicted in the diagram for clarity, but all correlations between factor variance components and covariates were freely estimated.

was interpreted as a low versus high *self-regulation factor that indicates vulnerability to general psychopathology (p)* versus *self-regulation capacity (src)*.

The second bipolar factor was positively loaded by social activity and negatively by social passivity, and it was interpreted as a *Social activity* factor. Social activity covered different types of activity regarding self-regulation: hyperactive and talkative behaviors as well as helping behavior and constructive problem solving. Correspondingly, passive behavior covered anxious as well as calm and patient behavior.

The fit indices showed that the RMSEA was close to the criteria and the SRMR achieved the criteria for a well-fitted model ($\chi^2(376) = 12142.61$, $p < 0.001$, $RMSEA = 0.086$, $CFI = 0.734$, $TLI = 0.692$, $SRMR = 0.071$). Large modification indices appeared in items belonging to subfactors (anxiety, aggression, hyperactivity or inattention), thus proposing a four factor model.

The four exploratory factors model fit the data better ($\chi^2(321) = 3484.95$, $p < 0.001$, $RMSEA = 0.048$, $CFI = 0.929$, $TLI = 0.903$, $SRMR = 0.032$), and the model fitted the data even better after freeing five residual covariances ($\chi^2(209) = 2532.84$, $p < .001$, $RMSEA = .041$, $CFI = .950$, $TLI = .931$, $SRMR = .028$).

**Table 1. Standardized factor loadings from exploratory factor analyses with two and four factors (MPNI Form SERI).**

| | Two factor model | | Four factor model | | | |
|---|---|---|---|---|---|---|
| | I | II | I | II | III | IV |
| *Hyperactive/Inattentive behavior* | | | | | | |
| **37** I am hyperactive | .54 | .51 | .06 | .68 | .09 | -.23 |
| **12** I talk all the time | .41 | .62 | .09 | .58 | .16 | -.37 |
| **3** I am restless and can't sit still | .58 | .23 | -.05 | .71 | -.01 | .06 |
| **24** I'm too impatient to wait for my turn | .59 | .17 | .13 | .54 | -.08 | .02 |
| **5** I act before thinking | .45 | .23 | .04 | .48 | -.06 | -.09 |
| **7** I am unable to concentrate on anything | .51 | -.04 | -.13 | .62 | -.05 | .32 |
| **14** I do not listen to directions | .57 | .08 | .07 | .51 | -.17 | .04 |
| **32** I forget things | .40 | -.03 | .00 | .40 | -.02 | .22 |
| *Aggressive behavior* | | | | | | |
| **33** I often become angry, and I easily get involved in quarrels or fights | .65 | .03 | .73 | .07 | -.06 | .04 |
| **36** When people yell at me, I yell back | .41 | .27 | .59 | .04 | .08 | -.21 |
| **11** I sometimes feel the desire to tease, to annoy, or to attack another person without reason | .49 | .02 | .42 | .14 | -.09 | .04 |
| **18** Given enough provocation, I may hit another person | .42 | -.01 | .68 | -.16 | -.11 | -.07 |
| **15** I spread rumors about other people's personal matters when I am mad at them | .42 | -.05 | .37 | .09 | -.04 | .10 |
| **8** When I am mad at someone, I sometimes decide to exclude him/her | .37 | -.01 | .48 | -.01 | .01 | .07 |
| *Anxious behavior* | | | | | | |
| **9** I usually do not feel at ease when I meet people I do not know too well | .21 | -.56 | .15 | -.05 | .01 | .61 |
| **16** I'm scared by and nervous about new things and situations | .16 | -.49 | .08 | .02 | .11 | .60 |
| **27** I am the kind of person who is excessively sensitive and easily hurt | .25 | -.31 | .15 | .11 | .17 | .57 |
| **35** Even though I know I am right I often have great difficulty getting my points across | .17 | -.34 | -.02 | .11 | -.00 | .48 |
| *Emotional self-regulation (reference variables for the two-dimensional model)* | | | | | | |
| **21** My moods change often, and I lose my temper easily | .58 | -.17 | .39 | .24 | .05 | .40 |
| **29** I am reliable and stable. I keep my composure in all situations | -.69 | .17 | -.23 | -.29 | .40 | -.10 |
| *Prosocial behavior (constructive and compliant behavior)* | | | | | | |
| **4** I am calm and patient | -.67 | -.10 | -.06 | -.54 | .31 | .09 |
| **28** I always do my tasks | -.51 | .08 | .07 | -.43 | .45 | .03 |
| **17** I'm a person everyone can trust | -.44 | .31 | -.03 | -.13 | .64 | .04 |
| **6** I try to solve difficult problems reasonably and consider other people | -.55 | .26 | .01 | -.32 | .59 | -.04 |
| **2** I am friendly to others | -.45 | .33 | -.18 | -.01 | .66 | .06 |
| **10** I sort out things through discussion | -.34 | .36 | -.14 | .04 | .52 | -.06 |
| **22** I help others when they need it | -.32 | .47 | .00 | .02 | .68 | -.01 |
| **13** I defend those who are weaker | -.15 | .44 | .01 | .14 | .52 | -.06 |
| *Social activity (reference variables for the two-dimensional model)* | | | | | | |
| **19** I'm very energetic, always on the go and often have contact with other people | .01 | .77 | .02 | .31 | .37 | -.56 |
| **34** I'm quiet, withdrawn, and often alone | -.03 | -.70 | .02 | -.25 | -.03 | .75 |

Note: In the two-factor model, the correlation between the factors (-0.02) was non-significant. In the four-factor model, the first factor correlated with the second, third and fourth factors by 0.56, -0.16 and 0.12, respectively. The second factor correlated with the third and fourth factors by -0.06 and 0.00, and the third factor with the fourth factor by -0.24. All non-zero correlations were statistically significant. In the four-factor model, residual correlations were 0.33 (#16 and #9), -0.27 (#15 and #17), 0.26 (#13 and #22), 0.33 (#4 and #29) and -0.27 (#21 and #29) due to item-level connections.

As seen in Table 1, low self-regulation divided into three factors that were positively loaded by low emotional regulation (#21), and that differed with respect to active versus passive behavior. Aggressive behavior (I) was independent of social activity vs passivity (#19 and 34), Hyperactive/Inattentive behavior (II) was loaded by activity, and Anxious behavior (IV) by passivity. High self-regulation was represented by Prosocial behavior (III), and high emotional regulation (#29) loaded positively only on this factor. Prosocial behavior was more highly loaded by social activity than passivity.

**Rating scales for self-regulation and invariance analysis of the scales across ages and informants**

Rating scales for factors obtained were created by using the mean scores of items identified. For the Self-regulation factor, two rating scales were formed, one for low and the other for high self-regulation, as follows: a *p-factor* scale was created from the 20 items of the MPNI Form SERI that loaded positively (showing low behavioral and emotional self-regulation, Table 1) on the first factor covering the reference variable for labile mood (#21). Correspondingly, an *src-factor* scale was created from the 9 items that loaded negatively on the first factor covering the reference variable for stable mood (#29). The reference variables for social activity (#19 and 34) were not included in these scales for behavioral and emotional self-regulation.

In addition, a Problem behavior scale was formed from the 19 behavioral items that loaded positively on the first factor (Table 1). Its subscales were formed and titled accordingly as *Hyperactive/Inattentive behavior*, *Aggressive behavior*, and *Anxious behavior*. For high self-regulation (8 items), a scale for *Prosocial behavior* was formed. The corresponding scales were formed for co-twin ratings at age 17, self- and co-twin ratings at age 14, teacher ratings at ages 12 and 14, and parent ratings at age 12.

Measurement invariance was tested on four levels: 1) Configural invariance - factor loadings and thresholds are freely estimated, 2) Metric (weak) invariance - factor loadings are set equal, and thresholds are freely estimated, 3) Scalar (strong) invariance - factor loadings and threshold are set equal, and 4) Residual (strict) invariance - factor loadings, thresholds are set equal [28] (see detailed test results in S1 Analyses).

Measurement invariance between girls and boys was tested separately for each informant. Strict invariance (equal factor loadings, thresholds and residual variances) held for teacher ratings at ages 12 and 14 and co-twin ratings at age 14. Partial strict invariance (2–5 item thresholds could not be set equal) held for other informants. These results meant that covariance and mean structure could be compared between boys and girls. Enriched data at age 14 were weighted to correspond to the proportion in the population.

Measurement invariance across informants was tested for each of the four factors separately, with only those items taken that theoretically belonged to the factor. Because weighted observations could not be used for enriched data at age 14, a sub-sample of enriched data was randomly selected that corresponded to the proportion in the population. Partial scalar invariance across informants and ages, indicating that both factor variances and the means of scales were comparable, held for Hyperactive/Inattentive behavior in both girls and boys and for Prosocial behavior in boys. In girls, partial scalar invariance for Prosocial behavior held across ages and informants in three invariance groups including a group for self- and co-twin ratings at ages 14 and 17. (See S1 Analyses, for the other groups in which the means of ratings can be compared.) For Anxious behavior, the level of invariance was metric for girls and boys. This means that only factor variances were comparable, not the means of the scales. For Aggressive behavior, partial scalar invariance in boys and girls held partly across informants: 1) across all informants at ages 12 and 14 and 2) across self- and co-twin ratings at age 17. Thus, the mean scores of the scales were not comparable between early and late adolescence. More detailed results are presented in S1 Analyses.

**Reliability and validity of the age 17 scales**

Reliability of the p-factor scale and the src-factor scale was equal and relatively high, and also equal for both sexes (Table 2). The presence of the emotional self-regulation items (#21 and #29) as the reference variables for self-regulation in the p-factor scale and the src-factor scale, respectively, slightly improved the reliability compared to the behavior scales.

The reliability of the co-twin ratings was similarly high. (For the formation and reliability of the MPNI's Social position and activity scale based on the items loading on the second bipolar factor called Social activity versus passivity, not included in the MPNI Form SERI, please see S2 Text).

The correlation between the *p-factor* and *src-factor* scales was -0.40 for females and -0.35 for males (Table 3), and these correlations were slightly higher than the correlations between Problem behavior and Prosocial behavior scales that did not include emotional regulation (-0.34 and -0.30, respectively). Additionally, the Problem behavior scale with its subscales correlated negatively with the Prosocial behavior scale (p<0.001). These correlations confirmed the *construct validity* of the MPNI Form SERI concerning the *p-* and *src-factor* scales. The Anxious behavior scale contributed to the p-factor less strongly than the Hyperactive/Inattentive and Aggressive behavior scales, and it was less independent of the *src-factor*.

Correlations of the MPNI Form SERI self-rating scales for self-regulation with social position items rated by the persons themselves and their co-twins (Table 4) showed some *criterion validity* of the scales with self-ratings for males and females. Popularity and leadership were more highly related to the src-factor scale than to the p-factor scale, whereas low resilience and victimization correlated more highly with the p-factor scale. Leadership and popularity (r=0.33 for both sexes) were also positively associated with p-factor activity (Hyperactive/Inattentive and Aggressive behavior scales). A

**Table 2. Cronbach's alphas of the MPNI Form SERI scales for self-regulation.**

|  | Self (age 17) | | Co-twin (age 17) | |
|---|---|---|---|---|
|  | **Females** | **Males** | **Females** | **Males** |
| *p-factor* scale | 0.79 | 0.79 | 0.82 | 0.84 |
| Problem behavior scale | 0.77 | 0.77 | 0.81 | 0.82 |
| Hyperactive/Inattentive behavior subscale | 0.76 | 0.73 | 0.81 | 0.81 |
| Aggressive behavior subscale | 0.69 | 0.68 | 0.77 | 0.76 |
| Anxious behavior subscale | 0.66 | 0.63 | 0.62 | 0.65 |
| *src-factor* scale | 0.78 | 0.78 | 0.85 | 0.86 |
| Prosocial behavior scale | 0.75 | 0.75 | 0.84 | 0.85 |

**Table 3. Spearman correlations\* of the MPNI Form SERI self-rating scales at age 17, separated by sex.**

|  |  | *p-factor* scale | Problem behavior scale | Hyperactive/ Inattentive subscale | Aggressive behavior subscale | Anxious behavior subscale | *src-factor* scale |
|---|---|---|---|---|---|---|---|
| Problem behavior scale | Females | **0.99** |  |  |  |  |  |
|  | Males | **0.99** |  |  |  |  |  |
| Hyperactive/Inattentive subscale | Females | **0.77** | **0.78** |  |  |  |  |
|  | Males | **0.81** | **0.82** |  |  |  |  |
| Aggressive behavior subscale | Females | **0.76** | **0.77** | **0.44** |  |  |  |
|  | Males | **0.76** | **0.77** | **0.45** |  |  |  |
| Anxious behavior subscale | Females | **0.43** | **0.41** | 0.02 | **0.08** |  |  |
|  | Males | **0.45** | **0.44** | 0.14 | 0.10 |  |  |
| *src-factor* scale | Females | **-0.40** | **-0.39** | **-0.34** | **-0.28** | **-0.15** |  |
|  | Males | **-0.35** | **-0.34** | **-0.31** | **-0.23** | **-0.18** |  |
| Prosocial behavior scale | Females | **-0.34** | **-0.34** | **-0.30** | **-0.24** | **-0.13** | **0.98** |
|  | Males | **-0.31** | **-0.30** | **-0.28** | **-0.20** | **-0.16** | **0.98** |

*Significant at p<0.001 if bolded.

closer inspection at the item level (not shown in Table 4) revealed that the correlations of leadership and popularity were highest with negotiating (#6 and #10) and helping (#13 and #22) behaviors for prosocial behavior, and that their positive correlations with the p-factor subscales were due to verbal activity: being talkative (#12 for hyperactivity) and verbal (not physical or indirect) aggression (#23 and #36). Victimization and low resilience (r = 0.20 for both sexes) were associated with problem behaviors, most highly with the Anxious behavior scale, particularly, with item 27 (excessively sensitive and easily hurt). For co-twin ratings of social position and resilience, the correlations were in the same direction but lower.

Additionally, for construct and criterion validity testing, Spearman correlations across different raters and ages were calculated for testing *concurrent* and *predictive validity* using the mean scores of the items for each scale for all informants at different ages and both sexes. The invariance analysis confirmed (see S1 Analyses) that the correlations of the Pro-social behavior scale and the subscales of Problem behavior can be statistically compared due to the metric invariance. All tested measurement model factor loadings could be set equal. The covariance/correlation structure can be compared between sexes and between informants.

The correlations ranged from low to moderate (all were significant at p < 0.05), across sex generally similarly (Table 5). The largest correlations were between self and co-twin ratings at age 17 (r range: 0.31-0.56) showing good *concurrent*

**Table 4. Spearman correlations\* of the MPNI Form SERI self-rating scales for self-regulation with social position and resilience items rated by self and co-twin, age 17.**

| | | *p-factor* scale | Problem behavior scale | Hyperactive/Inattentive subscale | Aggressive behavior subscale | Anxious behavior subscale | *src-factor* scale | Prosocial behavior scale |
|---|---|---|---|---|---|---|---|---|
| Self-rating items | | | | | | | | |
| Leadership (item #1) | Females | **0.05** | **0.07** | **0.14** | **0.20** | **-0.29** | **0.14** | **0.14** |
| | Males | **0.07** | **0.08** | **0.14** | **0.17** | **-0.27** | **0.25** | **0.26** |
| Popularity (item #26) | Females | -0.02 | -0.01 | **0.17** | **0.05** | **-0.35** | **0.23** | **0.23** |
| | Males | 0.03 | **0.05** | **0.15** | **0.13** | **-0.32** | **0.26** | **0.26** |
| Victimization (item #30) | Females | **0.19** | **0.18** | **0.09** | **0.11** | **0.21** | **-0.07** | **-0.06** |
| | Males | **0.24** | **0.23** | **0.15** | **0.14** | **0.26** | **-0.09** | **-0.08** |
| Low resilience (item #31) | Females | **0.32** | **0.30** | **0.15** | **0.11** | **0.43** | **-0.10** | **-0.07** |
| | Males | **0.28** | **0.27** | **0.16** | **0.11** | **0.39** | **-0.09** | **-0.07** |
| Co-twin rating items | | | | | | | | |
| Leadership (item #1) | Females | -0.02 | -0.01 | **0.07** | **0.08** | **-0.22** | 0.04 | 0.04 |
| | Males | **0.03** | **0.05** | **0.09** | **0.09** | **-0.17** | **0.16** | **0.17** |
| Popularity (item #26) | Females | 0.01 | 0.01 | **0.16** | 0.04 | **-0.24** | 0.07 | 0.08 |
| | Males | **0.06** | **0.07** | **0.15** | **0.09** | **-0.18** | **0.09** | **0.09** |
| Victimization (item #30) | Females | **0.10** | **0.10** | 0.04 | **0.07** | **0.11** | -0.02 | -0.02 |
| | Males | **0.10** | **0.09** | **0.08** | **0.05** | **0.07** | **-0.07** | **-0.07** |
| Low resilience (item #31) | Females | **0.21** | **0.20** | **0.11** | **0.08** | **0.26** | **-0.10** | **-0.08** |
| | Males | **0.13** | **0.12** | **0.12** | 0.02 | **0.13** | -0.04 | -0.03 |

*Significant at p < 0.05 if bolded.

validity of the ratings. Self-ratings at ages 17 and 14 (r range: 0.32-0.50), and co-twin ratings at ages 17 and 14 (r range: 0.36-0.54), across both sexes showed rather high *predictive validity* of the scales across 3 years. Among these correlation pairings, the p-factor, src-factor, and Hyperactive/Inattentive behavior scales showed the largest correlations (r range: 0.41-0.56), while Aggressive and Anxious behavior scales had smaller correlations (r range: 0.32-0.48 and 0.31-0.44, respectively). Correlations between different youth raters at different ages (self (age 17) with co-twin (age 14), or co-twin (17) and self (14)) showed lower predictive validity (r range: 0.19-0.41 and 0.19-0.39, respectively). Correlations between youth raters and adult raters (parents (age 12) and teachers (ages 12 and 14)) were the lowest (r range: 0.09-0.36).

### Twin analyses

Overall, MZ twin correlations were greater within pairs than were DZ twins (Table 6), suggesting significant genetic influences on these traits. Furthermore, correlations for female same-sex MZ and DZ pairs tended to be larger than correlations for male same-sex MZ and DZ pairs respectively, suggesting possible sex differences. Lastly, same-sex DZ twin correlations were generally larger than opposite-sex DZ twin correlations, also suggesting the presence of sex differences. (For twin correlations of co-twin ratings, see S2 Table).

In twin factor models, both the p-factor and src-factor were moderately heritable at ages 14 and 17 (A~0.5), and the emotional self-regulation covariates were modestly heritable (A~0.3) (Table 7, Fig 2). We did not identify significant shared environmental effects for either factor or covariate, suggesting that familial resemblance on these socioemotional behaviors at ages 14 and 17 is driven by additive genetic sources. Therefore, unique environmental sources made up the remainder of the variation in all four traits (E range: 0.40–0.78).

Table 5. Spearman correlations of the MPNI Form SERI scales for self-regulation between raters/ages, by sex.

| | | *p-factor* scale | Problem behavior scale | Hyperactive/ Inattentive subscale | Aggressive behavior subscale | Anxious behavior subscale | *src-factor* scale | Prosocial scale |
|---|---|---|---|---|---|---|---|---|
| Self (17)/Co-twin (17) | **Females** | 0.51 | 0.51 | 0.56 | 0.48 | 0.44 | 0.44 | 0.42 |
| | **Males** | 0.41 | 0.41 | 0.47 | 0.41 | 0.31 | 0.44 | 0.40 |
| Self (17)/Self (14) | **Females** | 0.47 | 0.47 | 0.47 | 0.32 | 0.39 | 0.47 | 0.47 |
| | **Males** | 0.45 | 0.45 | 0.50 | 0.34 | 0.32 | 0.47 | 0.49 |
| Self (17)/Co-twin (14) | **Females** | 0.33 | 0.33 | 0.35 | 0.21 | 0.26 | 0.30 | 0.32 |
| | **Males** | 0.30 | 0.29 | 0.41 | 0.19 | 0.17 | 0.30 | 0.30 |
| Co-twin (17)/Self (14) | **Females** | 0.33 | 0.32 | 0.39 | 0.26 | 0.26 | 0.32 | 0.31 |
| | **Males** | 0.36 | 0.36 | 0.37 | 0.25 | 0.19 | 0.30 | 0.30 |
| Co-twin (17)/Co-twin (14) | **Females** | 0.52 | 0.51 | 0.53 | 0.40 | 0.36 | 0.54 | 0.54 |
| | **Males** | 0.51 | 0.51 | 0.54 | 0.41 | 0.38 | 0.50 | 0.52 |
| Self (17)/Teacher (14) | **Females** | 0.14 | 0.15 | 0.25 | 0.13 | 0.17 | 0.16 | 0.16 |
| | **Males** | 0.25 | 0.25 | 0.36 | 0.20 | 0.15 | 0.19 | 0.19 |
| Self (17)/Teacher (12) | **Females** | 0.16 | 0.17 | 0.23 | 0.15 | 0.14 | 0.10 | 0.09 |
| | **Males** | 0.16 | 0.16 | 0.30 | 0.17 | 0.09 | 0.23 | 0.23 |
| Self (17)/Parent (12) | **Females** | 0.21 | 0.22 | 0.24 | 0.15 | 0.12 | 0.18 | 0.17 |
| | **Males** | 0.23 | 0.22 | 0.26 | 0.21 | 0.20 | 0.26 | 0.24 |
| Co-twin (17)/Teacher (14) | **Females** | 0.15 | 0.15 | 0.27 | 0.16 | 0.16 | 0.25 | 0.24 |
| | **Males** | 0.26 | 0.25 | 0.32 | 0.21 | 0.10 | 0.24 | 0.25 |
| Co-twin (17)/Teacher (12) | **Females** | 0.20 | 0.17 | 0.22 | 0.22 | 0.13 | 0.23 | 0.22 |
| | **Males** | 0.26 | 0.26 | 0.34 | 0.23 | 0.12 | 0.26 | 0.27 |
| Co-twin (17)/Parent (12) | **Females** | 0.29 | 0.22 | 0.28 | 0.23 | 0.23 | 0.26 | 0.26 |
| | **Males** | 0.23 | 0.22 | 0.30 | 0.23 | 0.10 | 0.32 | 0.30 |

PLOS Mental Health

**Table 6. Twin correlations for the MPNI Form SERI self-ratings at ages 14 and 17.**

| Scales/items | Age 14 | | | | |
| | MZ M | MZ F | DZ M | DZ F | OS DZ |
|---|---|---|---|---|---|
| Hyperactive/Inattentive | 0.46 [0.33, 0.57] | 0.57 [0.46, 0.66] | 0.29 [0.13, 0.43] | 0.26 [0.09, 0.42] | 0.36 [0.24, 0.46] |
| Aggressive behavior | 0.36 [0.22, 0.48] | 0.48 [0.36, 0.59] | 0.31 [0.15, 0.45] | 0.09 [-0.09, 0.26] | 0.03 [-0.1, 0.15] |
| Anxious behavior | 0.27 [0.13, 0.41] | 0.44 [0.31, 0.55] | 0.20 [0.04, 0.35] | 0.26 [0.09, 0.41] | 0.06 [-0.07, 0.18] |
| Prosocial behavior | 0.50 [0.38, 0.6] | 0.59 [0.49, 0.68] | 0.22 [0.06, 0.37] | 0.13 [-0.05, 0.3] | 0.11 [-0.01, 0.23] |
| Low emotion regulation | 0.45 [0.32, 0.56] | 0.52 [0.40, 0.62] | 0.37 [0.22, 0.50] | 0.43 [0.28, 0.57] | 0.12 [0, 0.24] |
| High emotion regulation | 0.13 [-0.02,0.28] | 0.27 [0.13, 0.4] | 0.16 [0, 0.31] | 0.08 [-0.1, 0.25] | 0.12 [0, 0.24] |
| | Age 17 | | | | |
| | MZ M | MZ F | DZ M | DZ F | OS DZ |
| Hyperactive/Inattentive | 0.49 [0.4, 0.57] | 0.50 [0.42, 0.58] | 0.27 [0.16, 0.37] | 0.15 [0.04, 0.25] | 0.11 [0.04, 0.19] |
| Aggressive behavior | 0.48 [0.38, 0.56] | 0.40 [0.32, 0.49] | 0.23 [0.13, 0.33] | 0.20 [0.10, 0.31] | 0.15 [0.07, 0.23] |
| Anxious behavior | 0.44 [0.35, 0.53] | 0.53 [0.45, 0.6] | 0.17 [0.06, 0.28] | 0.17 [0.06, 0.28] | 0.12 [0.04, 0.20] |
| Prosocial behavior | 0.47 [0.38, 0.56] | 0.54 [0.47, 0.61] | 0.31 [0.21, 0.41] | 0.21 [0.1, 0.31] | 0.20 [0.12, 0.27] |
| Low emotion regulation | 0.22 [0.11, 0.32] | 0.31 [0.21, 0.4] | 0.24 [0.14, 0.34] | 0.11 [0, 0.22] | 0.01 [-0.06, 0.09] |
| High emotion regulation | 0.33 [0.22, 0.42] | 0.38 [0.29, 0.47] | 0.15 [0.04, 0.25] | 0.01 [-0.10, 0.12] | 0.06 [-0.02, 0.14] |

As also indicated by the factor analysis within the twin model, the p-factor and src-factor were significantly negatively correlated at the phenotypic level at age 14 ($r_P = -0.47$) and 17 ($r_P = -0.42$). When decomposed into its sources of covariation, both the genetic and unique environmental correlations were also significant and negative at age 14 ($r_G = -0.65$, $r_E = -0.25$) and 17 ($r_G = -0.66$, $r_E = -0.20$). The p-factor and src-factor also covaried with the emotion regulation items in the expected directions.

## Discussion

This study created factors and respective scales for, and validated a measure of, socioemotional self-regulation assessed by self-ratings of adolescents with the MPNI. After a series of tentative factor analyses, two-factors and four-factors models were estimated using the age 17 MPNI Form SERI. The two-factors model consisted of two bipolar factors (one regarding self-regulation and one regarding social activity). The self-regulation factor was interpreted to depict low self-regulation indicating vulnerability to a p-factor versus high self-regulation indicating capacity for self-regulation (an src-factor). The four-factors model fitted the data better. Low self-regulation divided into three components: Aggressive behavior, Hyperactive/Inattentive behavior, and Anxious behavior. Low emotional regulation was common to these factors, but they differed in social activity, as depicted in Fig 1. High self-regulation was represented by Prosocial behavior loaded by high emotional regulation.

The subscales of behavioral self-regulation (excluding emotional regulation) were categorized into a Problem behavior scale (comprising Hyperactive/Inattentive, Aggressive, and Anxious behavior subscales) for low behavioral self-regulation and a Prosocial behavior scale for high behavioral self-regulation. With low emotional self-regulation (labile mood), the Problem Behavior scale formed the p-factor scale, whereas the Prosocial behavior scale with high emotional self-regulation (stable mood) formed the src-factor scale. The scales had moderate/high Cronbach's alphas and showed moderate/high construct, criterion, concurrent, and predictive validity.

Invariance analyses were conducted for factor loadings and covariance/correlation structures between sexes, ages, and informants. Both the factors and correlations were very similar. The implication of the invariance analysis was that factors and respective scales were highly comparable across informants (self, co-twin, teacher, parent). Correlations between the behavioral scales can be calculated using the sum or mean score of items for each scale. However, the sum or mean

**Table 7. Results of twin factor models at age 14 and 17.**

| Socioemotional behavior | Age 14 Variance Decomposition | | |
|---|---|---|---|
| | A | C | E |
| *p-factor* | 0.50 [0.28, 0.68] | 0.09 [0.00, 0.28] | 0.40 [0.30, 0.50] |
| *src-factor* | 0.52 [0.43, 0.59] | 0.01 [0.00, 0.07] | 0.47 [0.40, 0.54] |
| Low emotion regulation covariate | 0.41 [0.22, 0.54] | 0.08 [0.00, 0.23] | 0.52 [0.45, 0.59] |
| High emotion regulation covariate | 0.19 [0.03, 0.30] | 0.03 [0.00, 0.16] | 0.78 [0.70, 0.86] |
| | Correlations | | |
| Pair comparison | Phenotypic | Genetic | Unique Environmental |
| *p-factor – src-factor* | -0.47 [-0.52, -0.41] | -0.65 [-0.83, -0.58] | -0.25 [-0.36, -0.12] |
| *p-factor – low emotion regulation* | 0.75 [0.71, 0.78] | 0.96 [0.88, 0.99] | 0.50 [0.41, 0.59] |
| *p-factor – high emotion regulation* | -0.27 [-0.32, -0.21] | -0.78 [-0.99, -0.77] | -0.08 [-0.20, 0.03] |
| *src-factor – low emotion regulation* | -0.40 [-0.44, -0.36] | -0.69 [-0.91, -0.69] | -0.12 [-0.21, -0.03] |
| *src-factor – high emotion regulation* | 0.28 [0.23, 0.32] | 0.53 [0.52, 0.96] | 0.42 [0.10, 0.28] |
| low emotion regulation – high emotion regulation | -0.14 [-0.19, -0.10] | -0.58 [-0.99, -0.57] | 0.01 [-0.08, 0.09] |
| Socioemotional behavior | Age 17 Variance Decomposition | | |
| | A | C | E |
| *p-factor* | 0.51 [0.37, 0.63] | 0.03 [0.00, 0.13] | 0.45 [0.36, 0.55] |
| *src-factor* | 0.50 [0.40, 0.55] | 0.01 [0.00, 0.07] | 0.50 [0.45, 0.55] |
| Low emotion regulation | 0.31 [0.20, 0.37] | 0.01 [0.00, 0.08] | 0.68 [0.62, 0.75] |
| High emotion regulation | 0.31 [0.23, 0.37] | 0.01 [0.00, 0.06] | 0.68 [0.63, 0.74] |
| | Correlations | | |
| Pair comparison | Phenotypic | Genetic | Unique Environmental |
| *p-factor – src-factor* | -0.42 [-0.46, -0.38] | -0.66 [-0.81, -0.59] | -0.22 [-0.31, -0.13] |
| *p-factor – low emotion regulation* | 0.63 [0.59, 0.68] | 0.88 [0.80, 0.99] | 0.55 [0.47, 0.63] |
| *p-factor – high emotion regulation* | -0.56 [-0.60, -0.52] | -0.91 [-0.99, -0.71] | -0.38 [-0.45, -0.30] |
| *src-factor – low emotion regulation* | -0.23 [-0.26, -0.20] | -0.34 [-0.50, -0.19] | -0.16 [-0.23, -0.10] |
| *src-factor – high emotion regulation* | 0.53 [0.51, 0.56] | 0.72 [0.63, 0.81] | 0.42 [0.37, 0.47] |
| low emotion regulation – high emotion regulation | -0.43 [-0.45, -0.40] | -0.62 [-0.78, -0.49] | -0.33 [-0.39, -0.28] |

scores of the scales cannot be used for the comparison of the mean value of the scales across informants and ages, because it has not been done in the present study. Most of the models were also partially invariant when testing thresholds, therefore factor means can be compared between sexes but using latent factors. For the Problem behavior scale, one should consider whether the components for Hyperactivity/Inattentive, Aggression, and Anxiety need to be weighed differently at different ages and across informants. This is because variances are not comparable between ages and informants for this scale: using raw scores to sum the scales produces different weights for subscales than using standardized subscales.

The scales for low and high self-regulation at age 17 were shown to have high correlations with co-twin ratings at age 17, as well as similarly formed scales for self- and co-twin ratings at age 14. Due to the high reliability of the scales (0.7–0.8) and thus some remaining random measurement error, the real correlations were around 1.2 times higher than the observed correlations [29]. Significant predictive correlations were also found for the MPNI at age 12 and 14 for teacher and parent ratings. The twin models indicated that the p-factor and src-factor are moderately heritable and are influenced by both genetic and unique environmental factors. The factors correlate with each other and with covariates in the expected directions, and both genetic and unique environmental factors underlie the observed phenotypic correlations. The results were largely similar at ages 14 and 17.

The two-factors model was in accordance with the two-dimensional impulse control model (Fig 1) that describes socio-emotional behavior in terms of low versus high self-regulation and social activity versus passivity [8]. The four-factors model showed that low self-regulation divided into components for socially active, externalizing problems and socially passive, internalizing problems, as at age 11/12 MPNI. High self-regulation did not, however, divide into socially active and passive components as expected and as emerged at age 11/12; only prosocial behavior emerged at age 17. Possibly, a different item, such as "I keep my opinions to myself rather than express them publicly" would have indicated socially passive high self-regulation better at age 17 than the item #20 "I avoid difficult situations by doing something else" used at age 11/12 and repeated at age 17. More conceptual analysis and empirical work is necessary in the search for good indicators for high self-regulation expressed in a socially passive way. Extraversion is highlighted in Western cultures, while adaptive introversion is often ignored, even in research.

The MPNI scales are comparable to the SDQ regarding the deficit-oriented constructs (hyperactivity, conduct problems, emotional symptoms) and the strength-based construct (prosocial behavior) [30]. The similarity between the MPNI and SDQ shows a common aim to assess, on the one hand, a general psychopathology factor, p, and on the other, positive development, self-regulation capacity, src. The fourth deficit-oriented scale in the SDQ is peer problems, which resembles the scale that can be formed from the items for activity/passivity and social position (e.g., reverse coded popularity and leadership, and victimization) which loaded on the second factor in the tentative analysis of the age 17 MPNI and were dropped from the MPNI Form SERI because they did not assess self-regulation (see category "Others" in S1 Table and S1 Analyses).

A significant difference between the MPNI and SDQ is in their aims. The SDQ was constructed for clinical screening to differentiate those who need services, and the MPNI for the study of non-clinical samples. Goodman et al. [31] recommended using the five separate scales of the SDQ with clinical patients but using two scales, one for problem behaviors and the other for prosocial behaviors, with non-clinical samples due to the low frequency of clinical symptoms in the latter. The results of Staatz et al. [30] demonstrated that in the general population, the second-order factor model with one bipolar factor had a better fit and validity than a five-factor model. Both ends of the self-regulation dimension are predictive of future development of children, as shown by the JYLS study [8].

As expected for behavioral traits [32], MZ pairs were more strongly correlated for all scales compared to their DZ counterparts, indicating some underlying genetic variation. Similarly, the twin models indicated genetic and unique environmental influences underlying the factors identified by the factor analysis. The patterns of genetic and environmental correlations were in the expected directions and results were largely consistent between age 14 and 17.

Various other twin studies have evaluated the genetic and environmental influences underlying similar behaviors in adolescence. Our results are broadly in line with previous work suggesting both moderate genetic and unique environmental influences in prosocial behavior [1,33–36], anxiety [35,37,38], aggression [37,39,40], conduct and peer problems [35,36,41], attention deficit hyperactivity disorder (ADHD; 41,42), hyperactivity [35,43], broad externalizing psychopathology [44,45], and general psychopathology [46].

Future work in twin studies with the MPNI could evaluate longitudinal stability and change in underlying variation, variation shared with and unique to specific raters, and sex differences. It would also be important to study more closely the factors that affect the development of self-regulation of socioemotional behavior. In MPNI-assessed children, socioemotional behavior was previously associated with executive functions. Low inhibition and updating (containing working memory and shifting) explained low self-regulation (externalizing and internalizing problems), whereas high inhibition and updating explained high self-regulation (constructive and compliant behaviors) [8]. In adulthood, biological processes related to the p-factor, operationalized by externalizing and internalizing problems assessed with the MPNI, have been obtained by examining plasma proteomic and metabolomic profiles in young adults [47,48].

The level of self-regulation is most likely based on the developmental level of executive functions, temperament, and acquired skills during upbringing. Child-centered parenting (or lack of it) has been found to be associated with prosocial as well as problem behaviors [8]. At age 11/12, common environmental effects on teacher-rated socioemotional

behavior were significant [19], but at ages 14 and 17, shared environment did not explain self-rated socioemotional behavior. Possibly twins' experiences of parenting are so unique that their socioemotional behaviors are affected in unique ways. The effect of shared environment generally decreases until adolescence [49], but besides age, significant factors may lie in informants. As noted by Plomin, parents report only modest differential parenting of their children compared to the children's view, and perhaps shared environment explains teacher ratings more highly than self-ratings.

This study has many strengths, including multiple raters across developmental time points and a genetically informative, large and representative twin sample. However, it has limitations. First, the MPNI at age 17 did not include items for depression because a specific depression scale (GBI) was separately included in that collection wave. However, some proxy depression items, such as #27 (being excessively sensitive) from the anxiety scale of the age 17 MPNI Form SERI have been found to be associated with perceived occupational noise exposure, in a similar pattern as GBI [50]. Thus, aspect(s) of depression were captured in the age 17 MPNI. Social anxiety and depression jointly formed the factor for Emotional problems at age 12 [1]; depression items included "Worries a lot" and "Seems to be sad and depressed a lot of the time", which could be included in the age 17 MPNI Form SERI to create a factor for Emotional problems with anxiety. Another limitation is that we did not utilize a longitudinal study design to assess developmental trajectories; this was outside the aims and scope of this already ambitious study. Lastly, factor analyses suggested strong similarity for males and females on the composition of the factors, but sex- and zygosity-stratified twin correlations did offer some evidence of sex differences underlying the variation; however, to pursue this was not an aim of the present work.

## Conclusions

The MPNI Form SERI was shown to be a reliable and valid measure of low and high self-regulation of socioemotional behavior in early and late adolescence, and it can be used for self-ratings and those of co-twins or siblings. Our results were very similar for both sexes. Problem behavior can be complemented with items for depressive symptoms to create, with social anxiety, a more comprehensive sub-scale for emotional problems. Both low and high self-regulation of socioemotional behavior are heritable and develop from both genetic and unique environmental sources. The measure is well-suited for population studies.

## Supporting information

**S1 Text. MPNI age 17 factors, scales, and subscales derived from the current factor analysis (that comprise the MPNI Form SERI [socioemotional regulation inventory]) with the corresponding items in the MPNI age 12/14 questionnaires to create corresponding factors, scales, and subscales.**
(DOCX)

**S2 Text. Results on the scale for social position and activity (not included in the MPNI Form SERI).**
(DOCX)

**S1 Fig. Path diagram of basic twin model for a generic trait.** Here A stands for additive genetic component, C stands for shared environmental component, and E stands for unique environmental component, MZ refers to monozygotic or identical twins and DZ refers to dizygotic or fraternal twins.
(DOCX)

**S1 Table. Tentative results of the main dimensions of self-regulation of the age 17 MPNI\*: Factor analysis, orthogonally rotated factors (N = 4105); items grouped according to the components obtained for low and high self-regulation separately.**
(DOCX)

**S2 Table. Twin correlations for co-twin ratings at ages 14 and 17.**
(DOCX)

**S1 Analyses. Invariance analyses (text and tables).**
(DOCX)

## Acknowledgments

FinnTwin12 wishes to thank all participating twins and their parents and teachers.

## Author contributions

**Conceptualization:** Lea Pulkkinen, Jaakko Kaprio, Alyce M Whipp.

**Data curation:** Asko Tolvanen, Stephanie Zellers, Alyce M Whipp.

**Formal analysis:** Asko Tolvanen, Stephanie Zellers, Alyce M Whipp.

**Funding acquisition:** Jaakko Kaprio, Richard J. Rose.

**Investigation:** Lea Pulkkinen, Stephanie Zellers, Alyce M Whipp.

**Methodology:** Lea Pulkkinen, Asko Tolvanen, Stephanie Zellers, Alyce M Whipp.

**Project administration:** Lea Pulkkinen, Alyce M Whipp.

**Resources:** Jaakko Kaprio.

**Supervision:** Lea Pulkkinen.

**Validation:** Asko Tolvanen.

**Visualization:** Stephanie Zellers, Alyce M Whipp.

**Writing – original draft:** Lea Pulkkinen, Stephanie Zellers, Alyce M Whipp.

**Writing – review & editing:** Lea Pulkkinen, Asko Tolvanen, Stephanie Zellers, Jaakko Kaprio, Richard J. Rose, Alyce M Whipp.

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
