## [Decision Letter · Decision Letter 0]

22 Jun 2025

PMEN-D-25-00146

Self-regulation of socioemotional behavior in twin adolescents: structural validation of a multidimensional inventory

PLOS Mental Health

Dear Dr. Whipp,

Thank you for submitting your manuscript to PLOS Mental Health. After careful consideration, we feel that it has merit but does not fully meet PLOS Mental Health’s publication criteria as it currently stands. Therefore, we invite you to submit a revised version of the manuscript that addresses the points raised during the review process.

As you can see, the Reviewer 2 suggests some changes to your manuscript. After considering the review and reading the paper myself, I offer a number of points to address in a potential revision, as well.

   1. Please clarify the use of term “bi-factor”. Bifactor models, which are ubiquitous in the structural modeling of psychopathology and regulatory capacities, include superordinate general factors as well as subordinate specific factors. It is not clear which model is used in your study.

References

Clark, D. A., Hicks, B. M., Angstadt, M., Rutherford, S., Taxali, A., Hyde, L., ... & Sripada, C. (2021). The general factor of psychopathology in the adolescent brain cognitive development (ABCD) study: A comparison of alternative modeling approaches. Clinical Psychological Science, 9(2), 169-182.

Markon, K. E. (2019). Bifactor and hierarchical models: Specification, inference, and interpretation. Annual review of clinical psychology, 15(1), 51-69.

Snyder, H. R., Friedman, N. P., & Hankin, B. L. (2021). Associations between task performance and self-report measures of cognitive control: Shared versus distinct abilities. Assessment, 28(4), 1080-1096.

   2. Please present the figure depiciting the MPNI structural model described in the first aim.

   3. Please present a rationale for separating low and high self-regulation factors.  

   4. I specifically encourage you to consider your statistical approach. The hypothesised structure of the MPNI self-ratings at age 17 is presented at the end of the Introduction, therefore, I would expect to see it tested by structural equation modelling procedures. Even more so that you test measurement invariance later.

   5. The Analysis subsection describes measurement invariance testing. It might be helpful to present the findings in the main manuscript and include assessment of model fit for the total sample.

We look forward to receiving your revised manuscript.

Kind regards,

Helena R. Slobodskaya, M.D., Ph.D., D.Sc.

Academic Editor

PLOS Mental Health

Journal Requirements:

1. We have noticed that you have uploaded Supporting Information files, but you have not included a list of legends. Please add a full list of legends for your Supporting Information files after the references list. 

Additional Editor Comments (if provided):

Reviewers' comments:

Reviewer's Responses to Questions

**Comments to the Author**

1. Does this manuscript meet PLOS Mental Health’s publication criteria ? Is the manuscript technically sound, and do the data support the conclusions? The manuscript must describe methodologically and ethically rigorous research with conclusions that are appropriately drawn based on the data presented.

Reviewer #1: Yes

2. Has the statistical analysis been performed appropriately and rigorously?

Reviewer #1: Yes

3. Have the authors made all data underlying the findings in their manuscript fully available (please refer to the Data Availability Statement at the start of the manuscript PDF file)?

Reviewer #1: Yes

4. Is the manuscript presented in an intelligible fashion and written in standard English?

Reviewer #1: Yes

5. Review Comments to the Author

Reviewer #1: Title: Self-regulation of socioemotional behavior in twin adolescents: structural validation of a multidimensional inventory

Strengths

1. The study is grounded in Pulkkinen’s well-established two-dimensional impulse control model, which integrates cognitive control and behavioral expression. This provides a robust framework for interpreting the findings.

2. The distinction between self-regulation (goal-directed behavior) and self-control (conflict resolution) aligns with contemporary literature (e.g., Inzlicht et al., 2021), adding conceptual clarity.

3. The use of a large, longitudinal twin sample (FinnTwin12) enhances generalizability and allows for genetic and environmental analyses.

4. The Multidimensional Peer Nomination Inventory (MPNI) was adapted for self- and co-twin ratings, demonstrating flexibility in measurement across developmental stages.

5. Factor analyses (including bi-factor modeling) and invariance testing were thorough, supporting the structural validity of the proposed scales (e.g., *p*-factor for psychopathology, src-factor for self-regulation capacity).

6. High Cronbach’s alphas (0.7–0.8) for the *p*-factor and src-factor scales indicate strong internal consistency.

7. Convergent validity was demonstrated via correlations between self- and co-twin ratings, while predictive validity was shown across ages (12–17) and informants (parents, teachers).

8. Twin modeling revealed moderate heritability (~50%) for both *p*-factor and src-factor, consistent with broader behavioral genetics literature (Polderman et al., 2015). The lack of shared environmental effects at ages 14–17 aligns with developmental theories emphasizing unique environmental influences in adolescence.

9. The MPNI’s utility in population-based studies is highlighted, particularly for assessing both strengths (prosocial behavior) and difficulties (hyperactivity, aggression, anxiety).

Limitations and Critiques

1. The omission of depression items (due to separate GBI administration) weakens the *p*-factor’s comprehensiveness, as depression is a core internalizing symptom. The authors acknowledge this but argue that anxiety items partially capture emotional problems.

2. The src-factor’s focus on prosocial behavior neglects socially passive high self-regulation (e.g., introverted compliance). The suggested addition of items like “I keep opinions to myself” is valid but untested.

3. The bi-factor model’s interpretation of *p*-factor as “general psychopathology” conflates low self-regulation with psychopathology. While related, these constructs are not identical (e.g., some high-risk behaviors may reflect context-specific dysregulation rather than psychopathology).

4. The src-factor’s single-factor structure (prosocial behavior) lacks differentiation between socially active/passive regulation, unlike the age 11/12 MPNI. This may reflect developmental changes or measurement limitations.

5. Predictive correlations with adult informants (parents, teachers) were modest (r = 0.09–0.36), suggesting rater-specific biases or developmental shifts. The study does not disentangle these effects.

6. The absence of longitudinal genetic analyses (e.g., cross-lagged twin models) limits insights into how genetic/environmental influences evolve from childhood to adolescence.

7. While measurement invariance was confirmed, sex differences in twin correlations (e.g., higher female MZ correlations for anxiety) were not explored. This could reflect gendered socialization or genetic effects.

8. The sample’s homogeneity (Finnish twins) raises questions about cross-cultural applicability, particularly for prosocial behavior, which may vary by cultural norms.

9. The MPNI’s non-clinical focus limits direct comparability to tools like the SDQ, which screens for clinical thresholds. The authors’ suggestion to use aggregate scales (problem/prosocial) in non-clinical samples is pragmatic but untested in clinical populations.

Recommendations

o Include depression items in future MPNI versions to strengthen the *p*-factor.

o Test items capturing socially passive high self-regulation (e.g., emotional suppression) to refine the src-factor.

o Differentiate low self-regulation from psychopathology conceptually (e.g., by linking *p*-factor to specific outcomes like academic failure or substance use).

o Apply cross-lagged models to examine causal pathways between self-regulation, environment, and outcomes.

o Compare genetic/environmental influences across informants (e.g., why teacher ratings show higher shared environment effects).

o Replicate findings in diverse populations to assess cultural validity.

o Explore sex-specific genetic/environmental mechanisms (e.g., via sex-limitation models).

Conclusion

This study makes significant contributions to the assessment of adolescent socioemotional behavior by validating a theoretically grounded, multidimensional inventory (MPNI SERI). Its strengths include robust factor structure, genetic insights, and developmental validity. However, limitations in measurement breadth, clinical applicability, and cultural generalizability warrant further research. The MPNI shows promise for population-based research but may require adaptations for clinical or cross-cultural use.

6. PLOS authors have the option to publish the peer review history of their article (what does this mean? ). If published, this will include your full peer review and any attached files.

**Do you want your identity to be public for this peer review?** For information about this choice, including consent withdrawal, please see our Privacy Policy .

Reviewer #1: No

---

## [Editor Report · Decision Letter 1]

7 Sep 2025

PMEN-D-25-00146R1

Self-regulation of socioemotional behavior in twin adolescents: structural validation of a multidimensional inventory

PLOS Mental Health

Dear Dr. Whipp,

Thank you for submitting your manuscript to PLOS Mental Health. After careful consideration, we feel that it has merit but does not fully meet PLOS Mental Health’s publication criteria as it currently stands. Therefore, we invite you to submit a revised version of the manuscript that addresses the points raised during the review process.

Overall, the previous round of revisions strengthened the study including the structural analyses conducted and I am happy to inform you that, subject to minor revisions, your paper is suitable for publication. The revisions you need to make are as follows: 

Table 1. Standardized factor loadings for two and four exploratory factors: There are only two columns for factor loadings under the heading “Two factor model”. It is not clear where are factor loadings for the four-factor model.

Moreover, it is not clear whether Table 1 presents loadings from exploratory or confirmatory factor analysis.

We look forward to receiving your revised manuscript.

Kind regards,

Helena R. Slobodskaya, M.D., Ph.D., D.Sc.

Academic Editor

PLOS Mental Health

Journal Requirements:

1. We have noticed that you have uploaded Supporting Information files, but you have not included a list of legends. Please add a full list of legends for your Supporting Information files before or after the references list.
---

## [Editor Report · Decision Letter 2]

14 Sep 2025

Self-regulation of socioemotional behavior in twin adolescents: structural validation of a multidimensional inventory

PMEN-D-25-00146R2

Dear Dr. Whipp,

We are pleased to inform you that your manuscript 'Self-regulation of socioemotional behavior in twin adolescents: structural validation of a multidimensional inventory' has been provisionally accepted for publication in PLOS Mental Health.

Best regards,

Helena R. Slobodskaya, M.D., Ph.D., D.Sc.

Academic Editor

PLOS Mental Health